

# Attribution of surface ozone to NO$_x$ and VOC sources during two different high ozone events

Aurelia Lupaşcu[1], Noelia Otero[1,a], Andrea Minkos[2], and Tim Butler[1,3]

[1]Institute for Advanced Sustainability Studies (IASS), Potsdam, 14467, Germany
[2]German Environment Agency, 06844 Dessau-Roßlau, Germany
[3]Freie Universität Berlin, Institut für Meteorologie, Berlin, Germany
[a]now at: Oeschger Centre for Climate Change Research (OCCR), Bern, Switzerland

**Correspondence**: A.Lupascu (Aura.Lupascu@iass-potsdam.de)

**Abstrac**t. Increased tropospheric ozone (O$_3$) and high temperatures affect human health during heat waves. Here, we perform a source attribution that considers separately the formation of German surface ozone from emitted NO$_x$ and VOC precursors during two peak ozone events that took place in 2015 and 2018 which were associated with elevated temperatures. Results showed that the peak ozone concentrations can be primarily attributed to nearby emissions of anthropogenic NO$_x$ (from Germany and immediately neighboring countries) and biogenic VOC. Outside of these high ozone episodes, baseline ozone concentrations are attributed primarily to long-range transport, with ozone due to remote anthropogenic NO$_x$ emissions and methane oxidation adding to the tropospheric ozone background. We show that a significant contribution to modeled O$_3$ coming from German NO$_x$ or VOC emissions occurs mostly in southern Germany, emphasizing that the production of ozone depends on the local interplay among NO$_x$ and VOC precursors. Shipping activities in the Baltic and North Seas have a large impact on ozone predicted in coastal areas, yet a small amount of ozone from these sources can also be seen far inland, showing the importance of transported ozone on pollution levels. We have also shown that changes in circulation patterns during the peak O$_3$ episodes observed in Germany during the 2015 and 2018 heatwaves can affect the contribution of different NO$_X$ emission sources to total O$_3$, thus the possible influence of multiple upwind source regions should be accounted for when mitigation strategies are designed. Our study also highlights the good correlation between ozone coming from German biogenic VOC emissions and total ozone, although the diurnal variation in the ozone coming from biogenic sources is not dominated by the diurnal variation in biogenic emissions, and the peaks of ozone from biogenic sources are disconnected from local emission peaks. This suggests that the formation of O$_3$ from local German biogenic VOC emissions is not the sole factor that influences the ozone formation and other meteorological and chemical processes affect the diel variation of ozone having a biogenic origin. Overall, this study helps to understand the importance of a source attribution method to understand the sources of O$_3$ in Germany and can be a useful tool that will help to design effective mitigation strategies.

## 1 Introduction

Increased concentrations of ground-level ozone can harm humans and vegetation, especially during hot summer days (Sillmann et al., 2021). As shown by Analitis et al. (2014), the heatwave effect combined with the high ozone episodes is associated with an increase in mortality. Recent IPCC assessment reports (2012, 2018) associate an increase in the occurrence of extreme heatwave episodes with an increase in socioeconomic costs, and with an increase in ozone concentrations that cause morbidity from other diseases.



Several well-documented heatwaves associated with high ozone concentrations were registered over central and western Europe such as August 2003 (i.e., Vautard et al., 2005, Vieno et al., 2010) and July 2006 (i.e. Struzewska and Kaminski, 2008). Vautard et al. (2005) studied the 2003 heatwave using the CHIMERE model that reproduced

very well the high observed ozone concentration at European monitoring stations. Struzewska and Kaminski (2008) have identified three independent factors that influence the level of photochemical pollution, such as the circulation patterns, the geographical location, and the intensity of anthropogenic nitrogen oxides ($NO_x$) emissions. Many studies showed that the main meteorological factor which drives high ozone pollution episodes is the temperature (i.e., Lin et al., 2001, Jacob and Winner, 2009, Porter et al., 2015, Pusede et al., 2015, and references

therein; Otero et al., 2016, and references therein). Schnell and Prather (2017) showed that extreme events of ozone and temperature over North America often overlap, although consistent offsets in space and in time are occurring, most likely due to advection of emitted precursors.

During the last decades, global and regional chemical models were used to simulate the enhanced ozone concentrations associated with heatwaves (i.e., Guerova and Jones, 2007, Ordonez et al., 2010), as well as future

high ozone episodes (Shen et al., 2016). Meehl et al. (2018) used a model simulation for a present-day climate and showed that during heatwave days the surface ozone enhancement exceeds 10 ppb over Europe compared to non-heatwave days. Chemical transport models (CTMs) are an important tool for the simulation of transport and transformation of gases and to study the relationship between meteorology and high air pollution episodes. More recent developments of CTMs enabled their capacity to tag the emissions by source (e.g., Grewe et al., 2017,

Butler et al., 2018, 2020, Lupașcu and Butler, 2019, Mertens et al., 2020); hence they can help to identify the main precursor sources contributing to ozone peaks.

Several studies have quantified the contribution of long-range transport vs local emissions impact on summer peak ozone in Europe. Jonson et al. (2018) showed that, for most of the models used in their inter-comparison study except for the model CHASER_re1 in summer, the contribution to European seasonal mean ozone levels is larger

for regions from outside Europe than the contribution from European sources. Pay et al. (2019) showed that the imported ozone is the largest contributor to the ground level high ozone concentrations registered in Spain in July 2012. Lupașcu and Butler (2019) indicated that the local sources (precursors emitted within receptor regions) explain up to 35% of modeled daily maximum 8-hour average ozone (MDA8 $O_3$) in several receptor regions, while the remote sources accounted for 11% to 45% of the total surface ozone. Their work also showed the importance

of locally emitted NOx precursors for the ozone metrics that are especially influenced by high values of ozone, such as MDA8 $O_3$, AOT40, W126, and 95[th] percentile.

Methane ($CH_4$) oxidation explains a large part of the ozone formation in the troposphere. Wang and Jacob (1998) showed that the historic increase of global methane emissions and its subsequent oxidation led, among other causes, to the increase of tropospheric ozone concentrations, but this increase depends on the local availability of

OH. Turnock et al. (2018, 2019) showed that the reduction of $CH_4$ is a key factor for controlling the increase of future surface ozone. West and Fiore (2005) analyzed the advantages of low ozone due to methane emission reductions on agriculture, forest management, and human health and they found that the monetized global benefits would justify cutting the methane emissions by ~17%. In addition, Fiore et al. (2008) showed that the $CH_4$ oxidation contributes ~20% to the annual mean MDA8 $O_3$ at nearly all US MDA8 $O_3$ surface locations since the

pre-industrial era, while Butler et al. (2020) indicated that $CH_4$ oxidation explains 35% of their simulated global average tropospheric ozone burden.





Biogenic volatile organic compound (BVOC) emissions contribute strongly to the formation of ozone (e.g., Fiore et al., 2011, Curci et al., 2009, Qu et al., 2013, Tagaris et al., 2014, Zhang et al., 2017). Among them, isoprene is the main component. Biogenic isoprene emissions depend on multiple environmental drivers such as temperature, solar radiation, plant water stress, ozone, $CO_2$ concentrations, and land cover. Although Simpson et al. (1995) showed that during summertime the BVOC emissions exceed anthropogenic emissions in many countries, Simpson (1995) performed several simulations (one in which the isoprene emissions were turned off) and conclude that the isoprene emissions do not strongly influence the long-term average simulated $O_3$ concentrations. Using long-term measurements acquired at Ispra, Italy, Duane et al. (2002) however, showed that the contribution of locally emitted isoprene to the local ozone formation reached up to 50-75% during the summer of 2000. More recent studies that are mainly using two sets of simulation, one with and one without considering biogenic emissions, assessed the importance of BVOC on $O_3$ formation. Tagaris et al. (2014) showed that the European BVOC emissions increased the predicted MDA8 $O_3$ by 5.7% for July 2006. Jiang et al. (2019) have used two biogenic models, one predicting three times more isoprene emissions than the other, still the BVOC emissions contribute less than 10% of mean $O_3$ concentration in the summer of 2011. Churkina et al. (2017) indicate that biogenic emissions explain on average ~12% of modeled mean ozone in Berlin for two summers (June-August); yet, on hot summer days, these sources are responsible for up to 60% of the modeled MDA8 $O_3$ concentrations. Zhao et al. (2016) showed the use of a newer version of Model of Emissions of Gases and Aerosols from Nature (MEGAN) better reproduces the observed isoprene than the publicly available version of the MEGAN model integrated into WRF-Chem, which ultimately will impact the surface $O_3$ concentration. Moreover, Zhang et al. (2021) noted that the use of multiple versions of MEGAN has significantly influenced the ozone concentration due to the changes in simulated biogenic emissions.

One important sink for tropospheric ozone is given by ozone dry deposition, which is dominated by stomatal uptake by vegetation (Turnipseed et al., 2009, Lin et al., 2019, and references therein). Moreover, Lin et al. (2020) showed that a reduced ozone uptake by vegetation worsens the severe ozone pollution during heatwave events. When the water vapor pressure deficit is high, the plants close their stomata to prevent water loss. However, most models use the Wesely deposition scheme (Wesely, 1998) which does not account for humidity deficit and depends solely on temperature and radiation, and consequently, overpredicts the deposition of ozone under hot and dry conditions. Rydsaa et al. (2016) compared measurements of the stomatal conductance and stomatal ozone flux acquired during three summer field campaigns at three different sites in Italy with the modeled results predicted by using the Wesely scheme embedded within WRF-Chem. It has been shown that the model underestimated night-time stomatal ozone uptake, and consequently overpredicted ozone concentration in the stable night-time planetary boundary layer. The model also overestimated the daytime stomatal conductance, leading to underestimated midday ozone concentration, highlighting the existence of some systematic biases in the model parameterization.

In this study, we focus on the origin of elevated ozone levels observed over Germany during the 6 - 13 August 2015 and 1 - 8 August 2018 periods. The 2015 and 2018 summers have been the subject of several studies due to the severity in terms of temperature and precipitation. The summer of 2015 showed high drought conditions and was characterized by high temperatures in many parts of central and eastern Europe (Hoy et al., 2016, Ionita et al., 2016). Several heatwave episodes occurred from the end of June until September, which were linked to persistent blocking conditions and a northward deflection of Atlantic storm tracks (Ionita et al., 2016). Orth et al. (2016) showed that the 2015 summer was a concurrent dry and hot extreme in most of central Europe. Similarly, other





studies have analyzed the summer of 2018 in central Europe, which was characterized by extremely dry and hot conditions (Buras et al., 2020). Recently, Zscheischler and Fisher (2020) analyzed in more detail the compounding

impacts of hot and dry events in Germany during 2018 (March-November). They showed that 2018 was record-breaking both in high temperatures and low precipitation.

The main goal of this study is to determine the contribution of different precursor emission sources to the modeled ozone concentration in Germany during these two periods. For this purpose, we perform a source attribution of tropospheric $O_3$ to both $NO_x$ and volatile organic compound (VOC) precursors using the TOAST system

(Tropospheric Ozone Attribution of Sources with Tagging) as described by Butler et al. (2018) and implemented in WRF-Chem as described by Lupașcu and Butler (2019). Compared to previous studies, in which the impact of a certain emission source on total ozone concentration was simulated by switching on/off the emissions or using a source attribution of both $NO_x$ and VOC based on the chemical regime, our tagging methodology allows us to investigate the separate contribution of the anthropogenic $NO_x$ emitted in different European countries and regions

of the world and the contribution of anthropogenic and biogenic VOC emitted in several regions to ozone levels seen in Germany during the aforementioned periods.

We first assess the model's ability to reproduce the hourly observed meteorological parameters and trace gas concentrations at all measurement stations in Germany. Then, using source attribution, we provide information about the contribution of different precursors to the total ozone concentration.


## 2 Model simulation

## 2.1. Regional model description and set-up

In this study, the WRF-Chem model version 3.9.1 (Grell et al., 2005, Fast et al., 2006) was used to simulate the high ozone concentrations observed in Germany during the 2015 and 2018 heatwaves. The analyzed periods are

6-13 August 2015 and 1-8 August 2018. For this purpose, we have set up two nested domains, using a 25-km, and 5-km grid spacing for a coarser domain that covers Europe, and an inner domain that encompasses Germany. The vertical coordinates use 38-sigma stretched levels extended up to 20 km a.g.l., with a ~50 m grid spacing adjacent to the surface and 11 levels located within 3 km of the ground.

The physics options used for this study include the Morrison double-moment microphysics scheme (Morrison et

al., 2009), the Kain-Fritsch cumulus parameterization (Kain, 2004), the fast version of the Rapid Radiative Transfer Model (Iacono et al., 2008) for longwave and shortwave radiation. The surface and boundary layer schemes are the Noah-MP (Chen and Dudhia, 2001) and the Yonsei University PBL (YSU PBL) scheme (Hong, Noh, and Dudhia, 2006). Following the approach of Kuik et al. (2016), the CORINE dataset (EEA, 2014) was used instead of the default USGS land cover data set, as land use in Germany. In particular, the characterization

of urban areas is better captured by the CORINE dataset (see Kuik et al. 2016, and Churkina et al., 2017).

Initial and boundary conditions for the meteorological parameters are taken from the ECMWF reanalysis (Era-Interim dataset). To limit the divergence of large-scale flow compared to the observed synoptic conditions, we have nudged the temperature and wind above the boundary layer for the outer domain. Biogenic trace gas emissions are calculated online using the MEGAN model (Guenther et al., 2006). Anthropogenic emissions (based

on the year 2015) of CO, $NO_x$, $SO_2$, NMVOCs, PM10, PM2.5, and $NH_3$ are obtained from the TNO-MACC III emissions inventory (Kuenen et al., 2014). Based on Mailler et al. (2013) work that investigated the impact of



anthropogenic emission injection height in accurately simulating background $SO_2$, $NO_2$, and $O_3$ concentrations, we applied the European vertical emissions profiles from the plume rise model of Bieser et al. (2011).

In this work, the extended tagged MOZART chemical mechanism described in Lupaşcu and Butler (2019) is used
to track the ozone produced from $NO_x$ sources. We have also implemented within WRF-Chem a system that tracks ozone produced from VOC precursors, similar to Butler et al. (2018, 2020). As in Lupaşcu and Butler (2019), the tagged $O_3$ species are advected independently, and to correct the numerical errors associated with the advection scheme, we use a mass fixer in which the sum of all combined tagged tracers is set equal to the corresponding untagged concentration.

As in Butler et al. (2018, 2020), two simulations are performed, one with $NO_x$ tagging, and another with VOC tagging using otherwise identical emissions. As the VOC tagged configuration is substantially more computationally expensive than the $NO_x$ tagged configuration, we employ a reduced set of tags in the VOC-tagged simulation. The full set of tags (as used in the $NO_x$ tagged simulation) is shown in Table 1. This set of tags enables the simulated ozone in WRF-Chem to be attributed to emitted $NO_x$ from within and outside of Europe. Thirteen
regions are defined for tagging emissions within the WRF-Chem domain, and another four regions are defined outside of the WRF-Chem domain, with tagged emissions and ozone from these regions transported into WRF-Chem via the lateral boundaries (see below). In addition to these geographical tags, which are applied to anthropogenic emissions, we define four "global" tags (not associated with emissions from any specific geographical region) to track the ozone produced from biogenic and biomass burning emissions, lightning NOx,
and stratospheric input.

For the VOC tagged simulations, we no longer need the "lightning" tag, but instead we define an additional tag to account for methane as an ozone precursor. Instead of tagging anthropogenic VOC emissions with the full set of regions shown in Table 1, we use a reduced set of regions; the anthropogenic VOC emissions are tagged for two regions: Europe and the rest of the world. Instead of a single global tag for biogenic VOC emissions (as used in
the $NO_x$ tagged simulation), we define four regional tags: Berlin, the rest of Germany, the rest of Europe, and the rest of the world.

### 2.2. Chemical initial and boundary condition

As in Lupaşcu and Butler (2019), the boundary conditions for several HTAP2 source regions (Asia, North America, oceanic sources, and rest of the world), and natural types (biogenic, biomass burning, lightning
emissions, and stratospheric ozone) are obtained from the extended CAM-Chem version 1.2 simulations. For this purpose, we performed two model simulations using the extended $NO_x$ and VOC tagging mechanisms described in Butler et al. (2020) for 2015 and 2018. The HTAP_v2.2 emission inventory (Janssens-Maenhout et al., 2015) was used to provide anthropogenic emissions. Biomass burning emissions for 2015 and 2018 are taken from the FINN inventory (Wiedinmyer et al., 2011). Biogenic emissions of $NO_x$ and VOC are prescribed (Tilmes et al.,
190    2015).

### 3 Observations

Observations of ozone, $NO_2$, and NOx are provided by the German Environment Agency UBA (Umweltbundesamt), which collects hourly surface pollutant observations from several hundred stations located
across Germany, run by the Federal States. Also, the modeled 2 m temperature, mean sea level pressure, 10 m wind speed, and direction were compared with observations obtained from the German Weather Service (DWD)





(Kaspar et al., 2013). For this purpose, we used the observed meteorological data retrieved from the DWD ftp site (ftp://ftp-cdc.dwd.de/pub/CDC/observations_germany/climate/hourly/). Additionally, we have used 2m-temperature from the reanalysis ERA5 (Herbach and Dee, 2016) that provides hourly data on a regular latitude-longitude 0.25°x0.25° spatial resolution as a proxy for observed daily maximum 2m-temperature (T2MAX) at the air quality station locations.

## 4 Results

### 4.1 Evaluation of the regional model

Due to the strong links between the meteorological parameters and ozone concentrations in the atmosphere (see Coates et al., 2016, Otero et al., 2016), we have assessed the ability of the WRF-Chem model to simulate the temperature, pressure, relative humidity, and wind speed and direction temporal variability over Germany. To do so, we evaluated the modeled meteorological fields against observations using statistical scores including mean bias, normalized mean bias (NMB), index of agreement (IOA), and the correlation factor between simulated and measured values (r) (see Appendix 1).

Tables 2 and 3 show that the WRF-Chem model reproduces quite well the 2m temperature, surface pressure, and mean sea level pressure over Germany for both periods. On average, WRF-Chem overestimates observed surface pressure by less than 4.3 hPa and the mean sea level pressure by less than 0.8 hPa and reproduces very well the temporal evolution (r's > 0.92). The modeled 2m temperatures are on average close to those observed both in terms of predicted values (NMB's of -0.3%) and temporal evolutions (r > 0.84). At most of the observational sites, the nighttime 2m temperature is overestimated and the model underestimates the daytime temperature (Fig. 1). Therefore, the T2MAX is underestimated, on average, by 1.1° C in 2015 and 1.4° C in 2018, still lower than Wyszogrodzki et al. (2013), Chen et al. (2014), Karlický et al. (2020). The simulated relative humidity is fairly close to the observed one (NMB's of 14.5 and 20.1%, r's > 0.75). The relatively low NMB of wind direction suggests that the atmospheric flow is quite well reproduced. We note that the NMB of wind speed simulated by WRF-Chem is quite high (35.2% in 2015 and 31.6% in 2018). Previous work with the WRF-Chem model has also noted a high bias for simulated wind speed, especially when low wind speeds are observed (Fast et al., 2014, Gomez-Navarro, 2015, Solazzo et al., 2017, Kehler-Poljak et al., 2017, Gao et al., 2018). Tao et al. (2020) showed wind speeds' NMB of 54% at 5 km resolution as our setup. Gao et al. (2018) mentioned the errors in terrain data and reanalysis and relatively low horizontal and vertical resolution of the model among the factors contributing to overestimated wind speeds under calm wind conditions. Hence, we expect that horizontal mixing of ozone and emitted pollutants will be artificially enhanced in our simulations.

Tables 4 and 5 present the evaluation for the modeled trace gases at the "background" sites located in the inner domain for the 2015 and 2018 simulations. These "background" sites include "urban background", "suburban background" and "rural background". The model underestimates the observed concentrations and reproduces reasonably well the hour-to-hour variability of $NO_2$, and $NO_x$ during both analyzed periods. These discrepancies between modeled and observed $NO_x$ species could be related to 1) the errors associated with the predicted wind speed and direction which can lead to the displacement of modeled parcel of air relative to the observed values; 2) errors in the emissions inventory; 3) relatively low model resolution that could lead to underestimated emissions gradients as well as to increased diffusion of emission into grid cells and therefore the modeled grid cell concentration may not correctly represent the observed concentrations. Kuik et al. (2018) showed that during the



interval 06:00-17:00 UTC the traffic NOx emissions could be underestimated by a factor of 3 in the urban area of Berlin, so we might assume this could be also the case for other German cities. Several other studies, such as Tuccella et al. (2012), Pirovano et al. (2012), Georgiu et al. (2018), have also noted the overall tendency of
chemical transport models to underestimate the NOx concentrations at the European level.

The modeled O3 concentration is quite well reproduced (NMB's of 2.3% and -0.9% in 2015 and 2018). The comparison of hourly modeled and observed O3 at each measurement station (see also Fig.1) reveals a persistent modeled overestimate of nighttime concentration and an underestimation of midday concentrations. The errors in predicting accurately the meteorological inputs such as nighttime boundary layer height could also be associated
with the nighttime overprediction of surface ozone. Among the conditions that could lead to the underprediction of ozone peak value during the daytime, we could also point to the underestimation of T2MAX, uncertainties of emissions, both from anthropogenic (NOx) and biogenic sources, as well as an overprediction in wind speeds that will transport further downwind the locally emitted precursors, which ultimately leads to a reduced local O3 formation rate. During AQMEII-2 (Air Quality Model Evaluation International Initiative), the working groups
using WRF-Chem reported an overall underestimation of the summer O3 (Im et al., 2015). Tuccella et al. (2012) shows that, in August 2007, WRF-Chem tends to overestimate the low O3 concentrations and underestimated those situated at the high end of the O3 concentration distribution (see their Fig. 3).

The model is biased low when predicting surface MDA8 O3 (NMB's of -3.3% and -9.7% in August 2015 and August 2018, respectively). The temporal and spatial variability are well reproduced during these two events (r>
0.74 and IOA >0.73). Since the MDA8 O3 concentrations are influenced by ozone maxima values, the consistent underestimation of peak ozone values is also noted by several models (Tuccella et al., 2012, Im et al., 2015, Oikonomakis et al., 2018, Visser et al., 2019, Mertens et al., 2020). Visser et al. (2019) showed that, in July 2015, the European mean bias for MDA8 O3 was -14.2 µg/m3, while Oikonomakis et al. (2018) noted that the model consistently underestimates the afternoon (12:00-18:00 UTC) observed O3 values above 50 ppb for the summer of
2010. Moreoever, Tuccella et al. (2012) noted an MB for MDA8 O3 of -5 µg/m3 in 2007. Kryza et al. (2020) showed an NMB for MDA8 O3 of ~ -15% during the summers of 2017 and 2018.

In summary, the model simulated fairly well the diel and multiday variation of meteorological parameters and trace gases concentrations and slightly underestimates the peaks of temperature and ozone. Nevertheless, we acknowledge that a more comprehensive evaluation would also require the comparison between modeled and
observed isoprene and its oxidation products. These chemical variables play a crucial role in increasing O3 under summer conditions, but unfortunately they are not routinely measured. Expansion of routine air quality measurements to include biogenic VOCs or their oxidation products such as formaldehyde could significantly help with model evaluation.

**4.2 Characteristics of modeled and observed surface concentration**

In this section, we investigate the variability of O3 concentration levels under two different O3 peak events in Germany focusing on the 6 – 13 August 2015 and 1 – 8 August 2018 periods and provide a possible explanation on why, although the model captures well the mean O3 concentrations, it fails in reproducing the observed O3 peaks. Table 6 shows the percentage of stations where the daily MDA8 O3 target value of 120 µg/m3 was exceeded. Figures 2 and 3 depict the surface maps of daily observed and modeled MDA8 O3 concentrations during the high
ozone episodes. While the model captures the spatial and temporal observed pattern of MDAO3, it usually



underestimates the peak of MDA8 $O_3$ concentrations, as also shown in Table 6. This behavior is accentuated when the model fails to simulate the observed precipitations (not shown).

The synoptic situation during the analyzed periods is depicted in Fig. S1 and Fig. S2. Both years present similarities in their synoptic circulation patterns, although there are some small variations. In 2015, at the beginning of the

period, the weather in Germany was influenced by a relatively high-pressure field situated in a saddle formed by a ridge of the Azores Anticyclone and an anticyclone centered over NE Scandinavia. The Azores' ridge gradually extended towards the East was transformed into a blocking ridge over central Europe. This blocking pattern has two stagnant low-pressure systems on the edge, one in the North Atlantic and the other over North-East Africa. At the same time, the vertical structure was influenced by a ridge that favors the intrusion of warm tropical air coming

from Africa far to the north of Europe. These ridges formed a dome of high pressure that led to stagnant weather conditions characterized by low wind speed, strong insolation and subsidence, and high temperatures. Hence, the atmospheric subsidence associated with the anticyclonic field, leading to particularly stable atmospheric conditions, contributes to explaining the elevated ozone seen over southwest Germany in the first part of the period. As also shown in Otero et al. (2022), over Germany during the period 1999–2015, atmospheric blocking leads to

temperature anomalies > 5° C that induces MDA8 $O_3$ anomalies >20 µg/m³. An occluded front on August 9, 2015, led to reduced ozone values in West Germany, and it was followed by a warm front that affected North Germany, which explains the low ozone values observed/modeled in this region. In 2018, the observed ridge of the Azores High has gradually increased in magnitude and it became centered over the middle of the Atlantic. This high-pressure field has slowly retreated towards the West, thus the surface pressure influencing Germany became

shallow and favored the appearance of a cold frontal system and its moderate winds and cloudy/rainy weather that prevailed on 5 – 6 August 2018.

Using the scatter plots depicted in Fig. 4, we compare our modeled MDA8 $O_3$ and the daily T2MAX pair against the observed MDA8 $O_3$ and ERA5 T2MAX pair. We choose ERA5 T2MAX as a proxy for observed T2MAX since these measurements are not available at air quality station locations. We note that the modeled T2MAX and

MDA8 $O_3$ are relatively low correlated (r's of ~ 0.45); for both periods, the slopes of the linear fit to modeled MDA8 $O_3$ and T2MAX are comparable (3.24 µg m⁻³/°C and 4.04 µg m⁻³/°C). When the same analysis is applied to the observed MDA8 $O_3$ concentrations and ERA5 temperatures pair, we remark a relatively high correlation of these variables (r=0.76 and 0.62), a high ozone–temperature slope (5.6 and 5.2 µg m⁻³/°C) compared to the modeled ozone–temperature slope. As previously noted, the model failure in reproducing the meteorological variables is

not conducive to $O_3$ formation, and this could explain the model's inability to capture the observed MDA8 $O_3$ sensitivity to T2MAX. Clearly, the underestimation of T2MAX cannot be the only factor contributing to the differences in the modeled and observed MDA8 $O_3$-T2MAX slopes and intercept values, as +30 µg/m³ biases occur at T2MAX lower than 20° C. A scatter plot of modeled vs observed MDA8 $O_3$ and T2MAX (see Fig. S3) shows that, generally, the T2MAX is underestimated. While the observed values of MDA8 $O_3$ reach more than

120 µg/m³, the corresponding model values tend to be underestimated; for observed values below 75 µg/m³, the modeled values are generally overestimated. Pusede et al. (2015) has also shown that other meteorological factors, including the advection and vertical mixing, have an indirect impact on chemistry, hence on ozone production.

The accurate prediction of surface ozone concentration remains a challenge because the concentration depends not only on emissions but also on a detailed representation of physical and chemical processes. As in Rydsaa et al.

(2016), our Fig. 1 shows a low midday modeled ozone concentration when compared with observation, and high modeled night-time ozone. The stomatal resistance in the Wesely scheme is calculated using the first layer


(surface) temperature and solar radiation. Thus, the lasting high 2m temperature in 2018 (maximum of 34.3° C, and median of 21.8° C) compared to those modeled in 2015 (maximum of 36.1° C, and median of 19.6° C) leads to high stomatal resistance, consequently a reduced ozone uptake from vegetation that ultimately leads to an

increased modeled surface concentration. As noted in Section 4.1, several studies showed a systematic underestimation of WRF-Chem's peak $O_3$ values (Visser et al., 2019, Oikonomakis et al., 2018, Tuccella et al., 2012, Kryza et al., 2020). Stanier et al. (2021) noted that high MDA8 $O_3$ values were biased low as in our Fig. 4, while Lu et al. (2021) noted the MDA $O_3$ was in general underestimated in July 2018.

Moreover, as a potential effect of climate change, the high extreme temperatures in the future will hinder the $O_3$

control, thus knowing which sources of emissions are contributing to the severe $O_3$ pollution episodes will help to design effective emissions control.

**4.3 Influence of different emission sources on hourly ozone at individual stations**

In the following, we examine the impacts on $O_3$ concentrations in Germany of $NO_x$ and VOC precursors for 14 individual background stations, which are representative of the geographical distribution over the German states

(see Table 7). Figures 5 and 6 show the contribution of $NO_x$ precursors to surface ozone in 2015 and 2018, while Figures 7 and 8 show the contribution of VOC precursors during these periods.

The stratospheric contribution is similar in both simulations. Considering that the stratospheric ozone is advected within the domain through the lateral boundaries, the small differences in the stratospheric contribution noticed between the $NO_x$- and VOC-tagged simulations are related to the difference coming from CAM-Chem as explained

by Butler et al. (2018). The attribution of $O_3$ to different sources helps to identify the ozone from stratosphere-troposphere exchange. For both analyzed events, the daily variability revealed periods in which stratospheric ozone contribution can reach 70 µg/m$^3$. As an example, the intrusion of stratospheric ozone seen at the beginning of August 2018 in north Germany can be associated with the Icelandic low that reaches its minimum pressure and geopotential height on July 30, 2018, which led to a strong downward wind component, therefore an increase of

stratospheric ozone transported to the surface (see Fig. S2). As shown in Lupaşcu and Butler (2019), the surface stratospheric $O_3$ is attributed to the transport of stratospheric $O_3$ concentrations that originates from the lateral boundary concentrations (taken from the CAM-Chem extended model). Clockwise winds at the edge of the Azores High brought air pollutants such as stratospheric ozone and its precursors from Iceland and North-Atlantic to north Germany. The blocking situation impedes the westerly winds, therefore the transport of stratospheric ozone

decreased. Kalabokas et al. (2013) showed that summer ozone maxima in Cyprus are observed when rich-ozone air masses subside from the upper troposphere, and consequently, they transport stratospheric $O_3$. Similarly, our study shows that the stratospheric ozone may also have a large influence when the MDA8 $O_3$ peak values are observed for the northern regions of Germany. Yet, the prevailing southerly winds modeled in south Germany blocks the transport of stratospheric ozone in these regions.

**4.3.1 Attribution of ozone to NOx emissions**

Figures 5 and 6 show the German and remote contribution of $NO_x$ emissions to hourly $O_3$ at individual stations in different German states, as well as the observed $O_3$ concentrations and modeled wind speed and direction. The time series of observed $O_3$ concentrations were well captured by the model, although it fails in reproducing the high-end of observed $O_3$ values, as noted above. It can also be seen that the contribution of different $NO_x$ source

regions and source types to the ozone time series at each station varies greatly over time and space. The average





contributions of the global HTAP2 regions associated with long-range transport were ~20% to the total $O_3$ and remained relatively stable in both years. Emissions from lightning and biogenic NOx have the least contributions to total $O_3$ during these episodes.

For both episodes, the contribution of $NO_x$ German emissions to the ozone concentration is small most of the time

in the northern states, whereas the peak ozone events in the south-western and western German stations are mainly driven by German $NO_x$ sources, usually exceeding the total contribution of surrounding source regions. The peak ozone events in the southern states are also generally higher than the peak ozone events in the northern states. At the analyzed stations, a significant positive correlation between ozone from local $NO_x$ sources and total ozone is found in both years (r's of 0.57 in Bavaria to 0.89 in Thuringia in 2015 and 0.33 in Mecklenburg-Vorpommern to

0.84 in Saxony-Anhalt in 2018). This emphasized the importance of $NO_x$ German sources as a key factor that drives high levels of $O_3$. These results further revealed that the high contribution of $NO_x$ German sources to hourly $O_3$ is strongly connected with low wind speed values, endorsing the impact of high-pressure systems on serious local $O_3$ pollution. Furthermore, we note that high wind speeds bring polluted air from regions upwind of our stations' location.

The main $NO_x$ contributors to stations differ when we compare the $O_3$ pollution episodes in 2015 with those in 2018. As can be seen in Figs. 5 and 6, in 2015, the stations in Lower Saxony, North Rhine-Westphalia, Rhineland-Palatinate, Hesse, and Saarland are influenced by transport from sources in Benelux (up to 30 µg/m$^3$), France (up to 63 µg/m$^3$), and Italy-Switzerland (up to 22 µg/m$^3$) at the beginning of the analyzed period, while in 2018 the same source regions have a significant contribution to the same stations at the end of the analyzed period, up to

41, 57 and 29 µg/m$^3$, respectively. The prevailing winds that favor the influence of these source regions are from the southwest. If the dominant wind direction is from the south, the German sources can explain more than 50% of total $O_3$. Among stations in north Germany (Schleswig-Holstein, Mecklenburg-Vorpommern, Berlin, Brandenburg) we note a sporadic contribution from the Scandinavian Peninsula (SCA) and Baltic and North Seas (BNS) in both years.

The outflow of ozone from other European regions plays a significant role in ozone pollution episodes, as the wind direction shifts from one day to another. Central Europe (CEN) is responsible for a remarkably constant share of the total ozone in 2015 at stations in Berlin, Brandenburg, Saxony-Anhalt, Thuringia, Saxony, and Bavaria, while in 2018 it exhibits an erratic contribution to the total $O_3$ concentration (see Figs. 5 and 6). Among stations within Baden-Württemberg, Bavaria, and Saxony, we distinguish a noticeable influence of rest of the world (up to 32

µg/m$^3$), and RBT (Russia, Belarus, Ukraine, Turkey, Azerbaijan, Armenia, and Georgia) (up to 15 µg/m$^3$) in 2018, and of SEE (Bulgaria, Romania, Moldavia, Albania, Slovenia, Croatia, Serbia, Montenegro, Macedonia, Greece, and Cyprus) (up to 10 µg/m$^3$) in 2015. The large contribution of German and European sources during these events indicates that a reduction of high $O_3$ pollution could not be achieved without a regional collaboration of controlling emissions sources within Europe. However, we note that the transport of ozone and its precursors could be

exacerbated since our model does not capture the stagnant conditions and overestimates the wind speeds observed in calm wind conditions by more than 60% in both years.

Even though we identified the main source regions that could explain the origin of $O_3$ in different regions of Germany when the $NO_x$-tagged mechanism is employed, we note that there is a large variability within the same region. For example, in 2018, in Brandenburg, the contribution of the stratospheric ozone is higher in the northern

area of this region in comparison with the southern area (see Fig. S4). In Baden-Württemberg (Fig. S4), the stations




located in the east of the region are largely influenced by CEN (Central Europe), while those located westward exhibit the largest contribution from German sources and FRA (France).

Moreover, the simulations accounting for two pollution episodes occurring in two different years show that changes in circulation patterns between those two years can affect the contribution of different $NO_X$ emission sources to total $O_3$. Mitigation measures targeting anthropogenic NOx emissions for the reduction of ambient ozone should focus on widespread regional reductions rather than targeted local reductions.

Shipping activities in the Baltic and North Seas have also contributed to the hourly $O_3$ (see Figs. 5, 6). This is consistent with previous work (i.e., Lupaşcu and Butler, 2019, Pay et al., 2019, Aksoyoglu et al., 2016, Jonson et al., 2020) that highlighted the impact of shipping on ozone production near coastal regions. Apart from reinforcing the role of shipping on ozone predicted in coastal areas, our ozone attribution also shows that the inland regions, such as Bavaria and Baden-Württemberg are also impacted by ozone produced from ship's emissions (up to 3.7 and 4.7 µg/m$^3$, respectively, in 2015 and 7.2 and 9 µg/m$^3$ in 2018). This underlines the effect of transport of $O_3$ produced from possibly remote $NO_x$ emissions on total $O_3$ for mitigation purposes.

**4.3.2 Attribution of ozone to VOC emissions**

At the same stations, we investigate the impacts on $O_3$ concentrations in Germany of VOC precursors (see Figures 7 and 8). As in Butler et al. (2020), methane oxidation is one of the main contributors to the total ozone and it has an almost constant contribution throughout the analyzed periods, ranging from 23.3±5.8 to 31.2±5.5 µg/m$^3$ in 2015 and from 25.6±6.4 to 31.4±8.4 µg/m$^3$ in 2018. Most of the ozone from $CH_4$ as well as $CH_4$ itself is coming from the lateral boundaries. Even though we consider the domain-wide methane emissions in our system, we expect as in Butler et al. (2020b) that the highest share of ozone coming from methane to be attributed to intercontinental transport. European anthropogenic VOC sources contribute only modestly to the O3 concentration, with an average contribution of 9.3±5 µg/m$^3$ in 2015 and 7.9±4.7 µg/m$^3$ in 2018. Apart from the stratospheric intrusion event in 2018, most of the spatial and temporal variability in peak ozone is driven by the production of ozone from biogenic VOC emissions. The ozone attributed to biogenic VOC emissions also exhibits a geographical pattern, as for the $NO_x$-tagged simulations. The highest share of ozone from biogenic sources is modeled in the south and south-west of Germany, where peak ozone is also generally higher, as noted for $NO_x$ source attribution. This region is mostly covered by broadleaf deciduous trees (EEA, 2006), a vegetation class that has especially high isoprene emissions (Guenther et al., 2006, Pfister et al., 2008). The high temperatures and the large coverage of biogenic-emitting species in the afore-mentioned areas can explain the high levels of ozone in these regions. The contribution of biogenic VOCs to total $O_3$ is higher during the day, reflecting the onset of biogenic VOC emissions that combined with the high temperatures promotes the photochemical production of $O_3$ from $NO_x$ and VOC precursors. Comparison of high ozone events in Figs. 5 and 6 with the corresponding events in Figs. 7 and 8 show that biogenic VOC contribute to $O_3$ when they react with anthropogenic $NO_x$ from nearby sources.

As for $NO_x$-tagging, the ozone coming from German biogenic emissions correlates relatively well with the total ozone in both years (r's of 0.68 in Mecklenburg-Vorpommern to 0.88 in Lower Saxony in 2015 and 0.43 in Mecklenburg-Vorpommern to 0.80 in Hesse in 2018). Similar to $NO_x$ source attribution, we note that the outflow of ozone coming from biogenic sources from the rest of Europe has a considerable impact on ozone concentration seen at these stations. Moreover, at the analyzed stations, the ozone coming from biogenic sources outside Germany also displays a good correlation with the total ozone (r's of 0.48 in Mecklenburg-Vorpommern to 0.75 in North Rhine-Westphalia in 2015 and 0.15 in Saarland to 0.62 in 2018 in Thuringia). The low correlation between




ozone coming from the German $NO_x$ or VOC precursors and total ozone noticed mostly in Mecklenburg-Vorpommern is related to a low predicted photochemistry, as also indicated by T2MAX values (not shown).

The average contribution of ozone coming from German biogenic emissions ranges from 4.65% (4.1 µg/m³) for Mecklenburg-Vorpommern to 25.3% (26.1 µg/m³) for Baden-Württemberg in 2015 and from 5.6% (4.84 µg/m³)

for Mecklenburg-Vorpommern to 24% (23.2 µg/m³) for Baden-Württemberg in 2018, within the range of averaged contribution shown by Mertens et al. (2020). Mertens et al. (2020) used a source attribution method that attributes $O_3$ to all precursors, without distinguishing between $NO_x$ and VOC. They showed that during extreme ozone events the contribution of the land transport sector is gaining importance, and, more, the contribution of the biogenic emissions to ozone levels is increasing during these events. Their work indicates that 18.8±0.3% of their modeled

$O_3$ for the June-August 2008-2010 period could be explained by the large contribution of biogenic emissions (consisting of both biogenic VOC and soil $NO_x$). In contrast, our approach can separate the influence of these two distinct sources. At the analyzed stations we find that, on average, the soil $NO_x$ emissions explain just 7% of modeled $O_3$, while 37% of modeled $O_3$ is attributed to all European BVOC emissions, with 14% of ozone due to BVOC emissions just from Germany. These findings suggest that the biogenic emissions that contribute to the

high ozone levels seen in Mertens et al. (2020) are mainly attributed to the BVOCs, and not to the soil $NO_x$. Our focus on peak ozone episodes favors the enhancement of biogenic emissions, consequently of predicted $O_3$ attributed to those sources. Furthermore, Pay et al. (2019) used an $O_3$ source apportionment method that used all precursors in a single run and utilized the $H_2O_2/HNO_3$ ratio to determine if $O_3$ is VOC- or NOx-sensitive. They showed that their source sector which includes biogenic emissions could explain up to 8% of daily mean modeled

$O_3$ during days when the $O_3$ target values of 120 µg/m³ are exceeded in Spain. The relatively low contribution of biogenic sources to daily mean $O_3$ suggests that their region of interest was mostly in a $NO_x$-limited regime throughout the simulation. Moreover, their study highlighted the need of attributing the BVOCs to an individual source since these emissions represent 70% of VOC emissions in Spain. Our source attribution method allows us to investigate the contributions of these specific emission sources to ozone, and, in addition, we can separate

contributions from biogenic emissions of $NO_x$ or VOC precursors on ozone levels. Our method also allows a direct quantification of $O_3$ coming from BVOC in comparison with other studies in which the emissions of biogenic were simply turned on and off (i.e., Lee et al., 2014, Churkina et al., 2017, Sun et al., 2021).

Figure 9 shows the relative contribution of BVOC and its precursors to ozone peak events in the hotspot area of Baden-Württemberg where the average predicted ozone concentration is 115 and 102 µg/m³ in 2015 and 2018,

respectively. These results are averaged over each grid cell defined as Baden-Württemberg. The average contribution of ozone from German biogenic emissions is higher than 20 µg/m³ in both years (see Fig. 9). We note that the temporal variation of total $O_3$ and $O_3$ coming from German BVOC emissions are in good agreement (r's of 0.72 and 0.65). The German isoprene concentrations predicted in Baden-Württemberg build-up in the early morning and evening and are low during the daytime due to the reaction with OH that ultimately leads to ozone

formation in the area. Moreover, the ratio BVOC/$NO_x$ is higher during daytime (not shown) and it favors the production of $O_3$ by $NO_x$, leading to an enhanced $O_3$ formation from biogenic sources. On the other hand, Figure 9 also portrays that the diurnal variation in the ozone coming from biogenic sources is not dominated by the diurnal variation in biogenic isoprene emissions and the peaks of ozone from biogenic sources are disconnected from local emission peaks. These findings suggest that the formation of $O_3$ from isoprene oxidation takes place either in the

vicinity of the source or occurs in the regions downwind of Baden-Württemberg. Hence, the Baden-Württemberg biogenic VOC emissions are not the sole factor that influences the ozone formation and other meteorological and





chemical processes affect the diel variation of ozone having a biogenic origin. Furthermore, Fig. 9 depicts a relatively stable contribution of ozone coming from biogenic sources originating outside Germany (on average 18.6 and 13.8 μg/m³ in 2015 and 2018, respectively) to the baseline ozone, a consequence of the long ozone lifetime and the horizontal transport of ozone to Baden-Württemberg. Given the uncertainties in biogenic emissions estimates (Zhao et al., 2016, Zhang et al., 2021), a future study should include the use of MEGAN v2.1 as described by Zhao et al. (2016).

## 5    Conclusions

The WRF-Chem model was used to perform a source attribution of German surface ozone from emitted NO$_x$ and VOC during two peak ozone events that took place in 2015 and 2018. The results from our simulations demonstrate that the peak ozone concentrations, which are reached during episodes of high temperatures, can be primarily attributed to nearby emissions of anthropogenic NOx (from Germany and immediately neighboring countries) and biogenic VOC. Outside of these high ozone episodes, baseline ozone concentrations are attributed primarily to long-range transport, with ozone due to remote anthropogenic NOx emissions and methane oxidation adding to the tropospheric ozone background. Anthropogenic NMVOC emissions do not contribute significantly to peak ozone events, but rather make a modest contribution to the baseline ozone.

The attribution of modeled O$_3$ to NO$_x$ emissions at 14 stations in Germany showed that depending on the geographical location of stations, the nearby sources could have a significant contribution to ozone formation (mostly in south Germany), whereas in north Germany the high ozone concentrations are associated with enhancement of transported ozone or brought from aloft. Westerly stations are more prone to be also influenced by ozone transported from France and Benelux, while easterly stations show a constant share of O$_3$ transported from Central Europe, more pronounced in 2015, highlighting the importance of prevailing winds on ozone pollution levels at a given location. When attributing modeled O$_3$ to VOC emissions we find that the German biogenic emissions account for the largest fraction of ozone during these episodes, and they exhibit the same geographical pattern, as for NO$_x$ source attribution, stressing that the production of ozone depends on the local interplay among NO$_x$ and VOC precursors. Our study also highlights the good correlation between ozone coming from German biogenic emissions and total ozone, although the diurnal variation in the ozone coming from biogenic sources is not dominated by the diurnal variation in biogenic emissions, and the peaks of ozone from biogenic sources are disconnected from local emission peaks. This suggests that the formation of O$_3$ from local German biogenic VOC emissions is not the sole factor that influences the ozone formation and other meteorological and chemical processes affect the diel variation of ozone having a biogenic origin. Moreover, the relatively stable contribution of ozone coming from biogenic sources originating outside Germany to the baseline ozone underscores the combination of long lifetime and horizontal transport of ozone.

Shipping activities in the Baltic and North Seas have a great impact on ozone predicted in coastal areas, yet a small amount of ozone from these sources can also be seen far inland, stressing the importance of transported ozone on pollution levels. These findings complement those of previous studies, such as Erikson et al., 2021, and references therein, that showed that regional actions are needed to reduce the peak ozone concentration during high ozone episodes. We have also shown that changes in circulation patterns between different peak O$_3$ episodes observed in 2015 and 2018 can affect the contribution of different NO$_X$ emission sources to total O$_3$, as such same source regions significantly impact the O$_3$ concentration at a given location at the beginning of the analyzed period in



2015 and the end of the analyzed period in 2018. Thus, the possible influence of multiple upwind source regions should be accounted for when mitigation strategies are designed.

Overall, this study provides useful findings on how emissions from local and remote sources influence the predicted $O_3$ and MDA8 $O_3$ during two high ozone episodes. Biogenic VOC emissions as well as the NOx emitted

in nearby regions enhance the $O_3$ production during episodes of higher temperatures. Given the high importance of biogenic VOC in determining the peak ozone concentrations, the lack of VOC measurements for evaluation of the modeled VOCs is another source of uncertainty in modeled ozone production. Additional VOC measurements, including biogenic VOC are necessary to improve our understanding of how well the modeled ozone precursors are simulated, consequently, the total $O_3$ concentrations.

**Code and data availability.**

The WRF-Chem model is publicly available on http://www2.mmm.ucar.edu/wrf/users/download/get_source.html. The modification introduced and described in Section 2 as well as the model data can be provided upon request to the corresponding author.

**Appendix 1. Statistical scores**

The main statistical metrics employed in this study are Pearson correlation coefficient ($r$), Mean Bias (MB), Normalized Mean Bias (NMB), and Index of Agreement (IOA). These statistical indicators are defined as follows, with $n$ the number of model–observation pairs, $M$ the modeled values (with $\bar{M} = \frac{\sum_{i=1}^{n} M_i}{n}$ the averaged modeled value) and $O$ the observations (with $\bar{O} = \frac{\sum_{i=1}^{n} O_i}{n}$ the averaged observed value):

$$MB = \frac{1}{n} \sum_{i=1}^{n} (M_i - O_i)$$


$$NMB = \frac{\sum_{i=1}^{n} (M_i - O_i)}{\sum_{i=1}^{n} O_i}$$

$$r = \frac{\sum_{i=1}^{n} (M_i - \bar{M})(O_i - \bar{O})}{\sqrt{\sum_{i=1}^{n} (M_i - \bar{M})^2} \sqrt{\sum_{i=1}^{n} (O_i - \bar{O})^2}}$$

$$IOA = 1 - \frac{n * \sqrt{\frac{1}{n} \sum_{i=1}^{n} (M_i - O_i)^2}}{\sum_{i=1}^{n} ((|M_i - \bar{M}| + |O_i - \bar{O}|)^2}$$

**Author contributions**

AL and TB designed the research. AL performed the model runs. The analysis of model runs was performed by AL with input from each co-author. AL drafted the paper with contributions from all the co-authors.



**Acknowledgments**

We acknowledge the use of the WRF-Chem preprocessor tools (bio_emiss, fire_emiss, mozbc) provided by the Atmospheric Chemistry Observations and Modeling Lab (ACOM) of NCAR.

**Financial support**

This work was hosted by IASS Potsdam, with financial support provided by the Federal Ministry of Education and Research of Germany (FBMBF) and the Ministry for Science, Research and Culture of the state of Brandenburg (MWFK).

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


**Table 1**. List of tagged European source regions for the NOx tagging mechanism

| Category | Acronym | List of countries |
|---|---|---|
| European source regions | MBS | The Mediterranean, and Black Seas |
| | BNS | Baltic, and North Seas |
| | CEN | Austria, Hungary, Czech Republic, Slovakia, Estonia, Latvia, Lithuania, Poland |
| | ITS | Italy, Malta, and Switzerland |
| | SEE | Bulgaria, Romania, Moldavia, Albania, Slovenia, Croatia, Serbia, Montenegro, Macedonia, Greece, and Cyprus |
| | IBE | Spain, and Portugal |
| | UKI | The United Kingdom, and Ireland |
| | GER | Germany (without Berlin) |
| | BER | Berlin |
| | BNL | Belgium, Netherland, Luxembourg |
| | SCA | Finland, Norway, Sweden, Denmark, and Island |
| | FRA | France |
| | RBT | Russia, Belarus, Ukraine, Turkey, Azerbaijan, Armenia, and Georgia |
| HTAP2 | ASI | Chemical boundary condition of modeled species coming from Asia |
| | NAM | Chemical boundary condition of modeled species coming from North America |
| | OCN | Chemical boundary condition of modeled species coming from shipping activities |
| | RST | Chemical boundary condition of modeled species coming from rest of the world |
| Global | BIO | Biogenic |
| | BMB | Biomass burning |
| | LGT | Lightning |
| | STR | Stratospheric $O_3$ |





**Table 2.** Observed mean and simulation summary statistics for meteorological parameters. The bias, normalized mean bias (NMB), correlation coefficient (r), and index of agreement (IOA) are calculated between simulated and observed meteorological parameters at the DWD stations during the 6 – 13 August 2015 period.

| Variable name | Observed mean | Bias | NMB (%) | r | IOA |
|---|---|---|---|---|---|
| 2m temperature | 22.9 | -0.7 | -0.3 | 0.86 | 0.90 |
| T2MAX | 28.5 | -1.1 | -0.3 | 0.42 | 0.56 |
| Relative humidity | 65.7 | 9.5 | 14.5 | 0.65 | 0.76 |
| Pressure | 984.01 | 4.1 | 0.4 | 0.91 | 0.98 |
| Mean sea level pressure | 1017.1 | 0.8 | 0.07 | 0.96 | 0.96 |
| Wind speed | 2.9 | 1.0 | 35.2 | 0.47 | 0.64 |
| Wind direction | 178.29 | 11.1 | 6.6 | 0.47 | 0.73 |

**Table 3.** Observed mean and simulation summary statistics for meteorological parameters. The bias, normalized mean bias (NMB), correlation coefficient (r), and index of agreement (IOA) are calculated between simulated and observed meteorological parameters at the DWD stations during the 1 – 8 August 2018 period.

| Variable name | Observed mean | Bias | NMB (%) | r | IOA |
|---|---|---|---|---|---|
| 2m temperature | 23.9 | -0.9 | -0.2 | 0.84 | 0.86 |
| T2MAX | 30.4 | -1.4 | -0.5 | 0.22 | 0.28 |
| Relative humidity | 58.7 | 12.1 | 20.1 | 0.67 | 0.75 |
| Pressure | 984.5 | 4.3 | 0.4 | 0.91 | 0.92 |
| Mean sea level pressure | 1017.1 | 1.1 | 0,1 | 0.98 | 0.97 |
| Wind speed | 2.8 | 0.9 | 31.6 | 0.49 | 0.65 |
| Wind direction | 190.1 | 11.6 | 6.1 | 0.41 | 0.69 |

**Table 4.** Observed mean and simulation summary statistics for $NO_2$, $NO_x$, $O_3$ and MDA8 $O_3$ concentrations. The bias, normalized mean bias (NMB), correlation coefficient (r), and index of agreement (IOA) are calculated between simulated and observed concentrations at the German "background" stations during the 6 - 13 August 2015 period

| Variable name | Observed mean ($\mu g/m3$) | Bias | NMB (%) | r | IOA |
|---|---|---|---|---|---|
| $NO_2$ | 13.5 | -3.0 | -22.3 | 0.39 | 0.59 |
| $NO_x$ | 15.8 | -4.7 | -30.2 | 0.36 | 0.52 |
| $O_3$ | 101.4 | 2.3 | 2.3 | 0.69 | 0.79 |
| MDA8 $O_3$ | 128.7 | -4.3 | -3.3 | 0.74 | 0.82 |



**Table 5.** Observed mean and simulation summary statistics for $NO_2$, $NO_x$, $O_3$ and MDA8 $O_3$ concentrations. The bias, normalized mean bias (NMB), correlation coefficient (r), and index of agreement (IOA) are calculated between simulated and observed concentrations at the German "background" stations during the 1 – 8 August 2018 period

| Variable name | Observed mean ($\mu g/m3$) | Bias | NMB (%) | R | IOA |
|---|---|---|---|---|---|
| $NO_2$ | 12.8 | -1.7 | -14.0 | 0.43 | 0.65 |
| $NO_x$ | 14.9 | -2.5 | -16.5 | 0.37 | 0.61 |
| $O_3$ | 96.9 | -0.9 | -0.9 | 0.62 | 0.74 |
| MDA8 $O_3$ | 130.5 | -12.5 | -9.7 | 0.66 | 0.73 |


**Table 6.** Percentage of stations in each day which exceeds the daily MDA8 $O_3$ target value during 6 – 13 August 2015 and 1 – 8 August 2018 periods

| 2015 | | | | | | | | |
|---|---|---|---|---|---|---|---|---|
| Date | 06 Aug. | 07 Aug. | 08 Aug. | 09 Aug. | 10 Aug. | 11 Aug. | 12 Aug. | 13 Aug. |
| % of observed exceedances | 90 | 82 | 58 | 35 | 55 | 57 | 50 | 69 |
| % of modeled exceedances | 91 | 82 | 59 | 30 | 40 | 42 | 44 | 46 |
| 2018 | | | | | | | | |
| Date | 01 Aug. | 02 Aug. | 03 Aug. | 04 Aug. | 05 Aug. | 06 Aug. | 07 Aug. | 08 Aug. |
| % of observed exceedances | 49 | 80 | 92 | 73 | 19 | 49 | 89 | 58 |
| % of modeled exceedances | 48 | 60 | 66 | 41 | 9 | 15 | 63 | 48 |









**Table 7**. List of selected stations

| Acronym | State | Station code | Station name | Station type |
|---|---|---|---|---|
| BE | Berlin | DEBE062 | Berlin Frohnau | Background rural |
| BB | Brandenburg | DEBB086 | Blankenfelde-Mahlow | Background suburban |
| BW | Baden-Württemberg | DEBW013 | Stuttgart-Bad Cannstatt | Background urban |
| BY | Bavaria | DEBY093 | Sulzbach-Rosenberg/Lohe | Background suburban |
| HE | Hesse | DEHE022 | Wiesbaden-Süd | Background urban |
| MV | Mecklenburg-Vorpommern | DEMV026 | Garz | Background rural |
| NI | Lower Saxony | DENI052 | Allertal | Background suburban |
| NW | North Rhine-Westphalia | DENW081 | Borken-Gemen | Background rural |
| RP | Rhineland-Palatinate | DERP022 | Bad Kreuznach | Background urban |
| SH | Schleswig-Holstein | DESH015 | Itzehoe | Background suburban |
| SL | Saarland | DESL019 | Biringen | Background rural |
| SN | Saxony | DESN059 | Leipzig-West | Background suburban |
| ST | Saxony-Anhalt | DEST104 | Domäne Bobbe | Background rural |
| TH | Thuringia | DETH020 | Erfurt | Background urban |

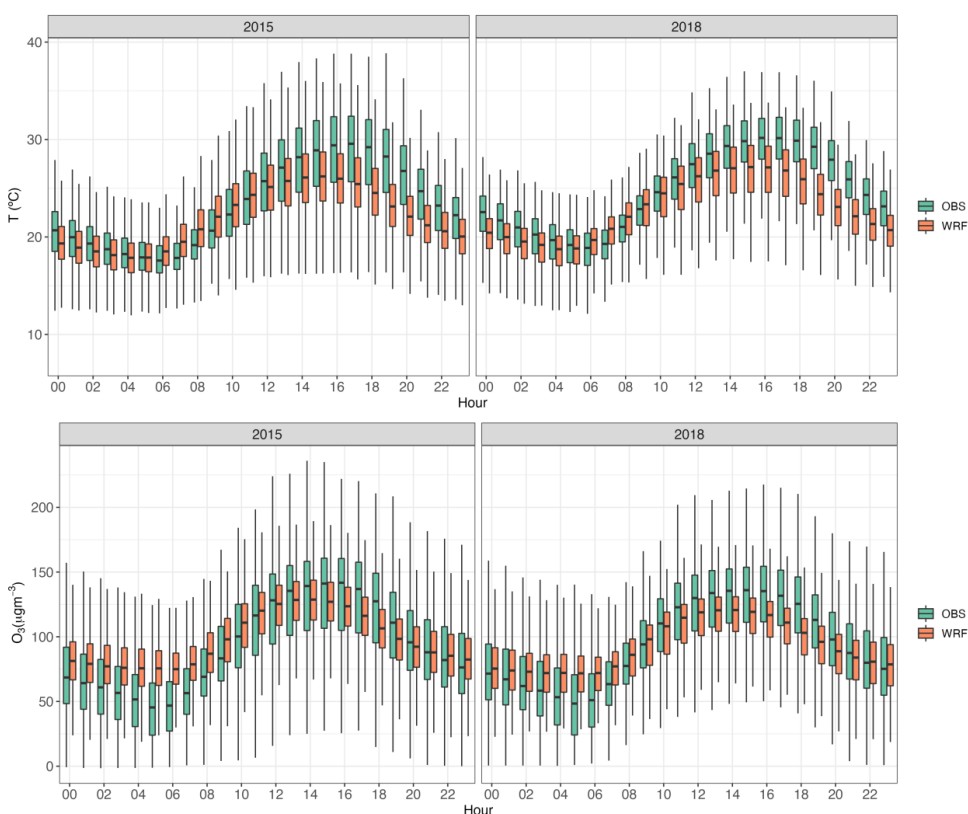

**Figure 1** Observed and simulated diel percentile for temperature and $O_3$ over the analyzed periods at the German "background" stations. Note that the "OBS" temperature is given by ERA5 temperature extracted at the German background station locations. Vertical lines denote 5[th] and 95[th] percentiles, boxes denote 25[th] and 75[th] percentiles, and the black line denotes the 50[th] percentiles



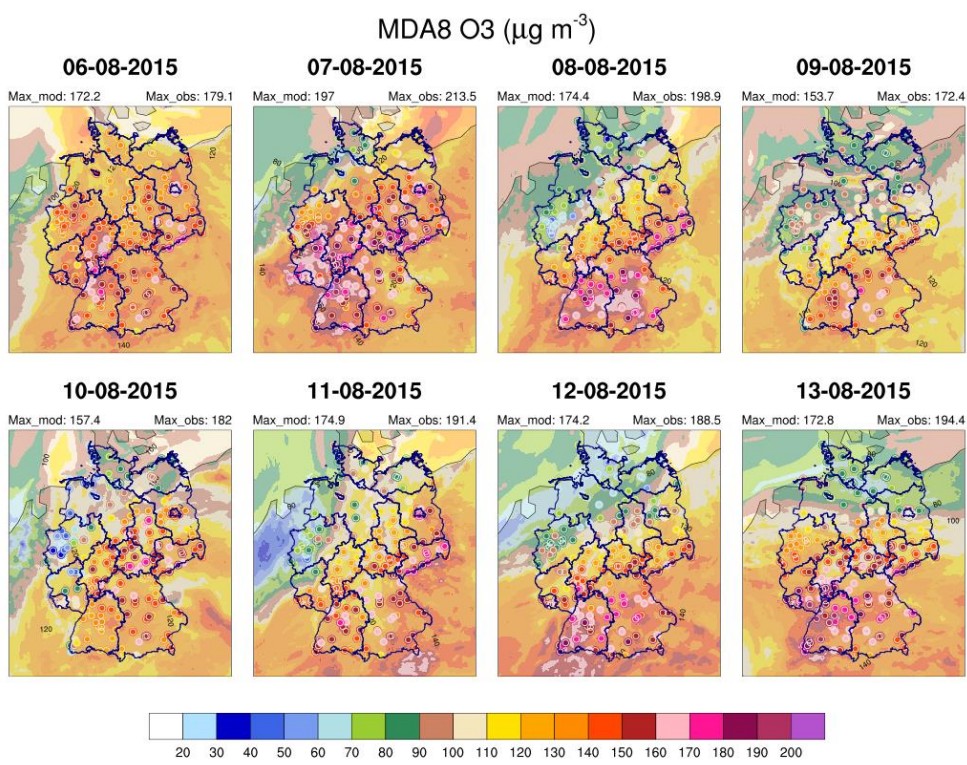

**Figure 2**. Daily surface MDA8 $O_3$ (µg/m³) for the 6 – 13 August 2015 period. Color dots represent the observed MDA8 $O_3$ concentrations at the German "background" stations.

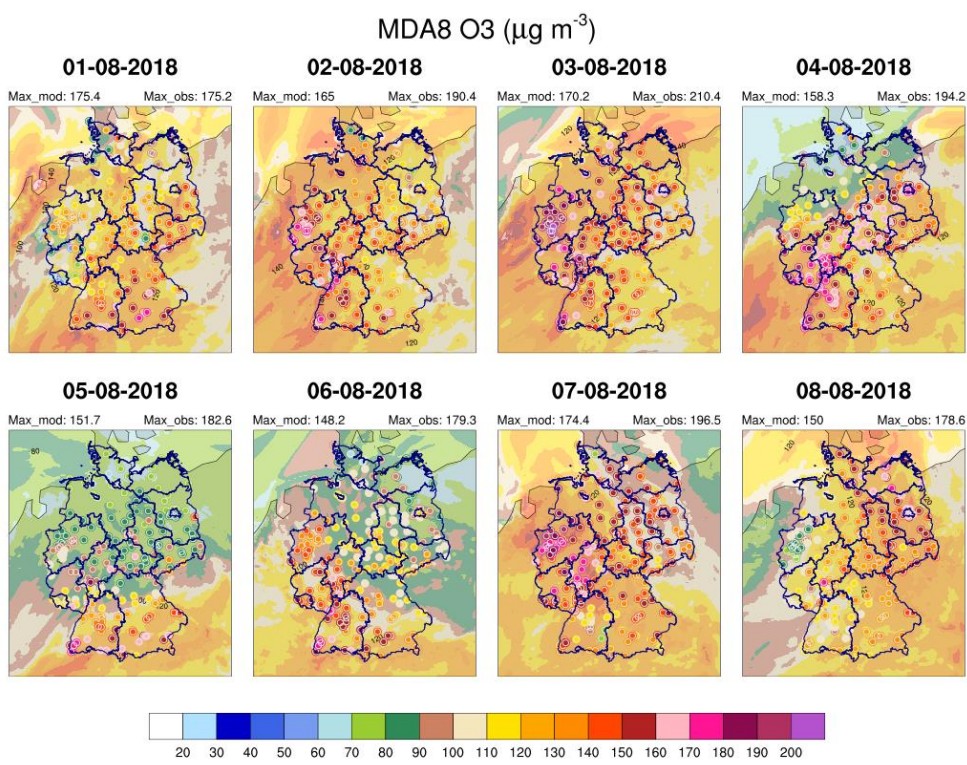

**Figure 3.** Daily surface MDA8 $O_3$ (µg/m³) for the 1 – 8 August 2018 period. Color dots represent the observed MDA8 $O_3$ concentrations at the German "background" stations.





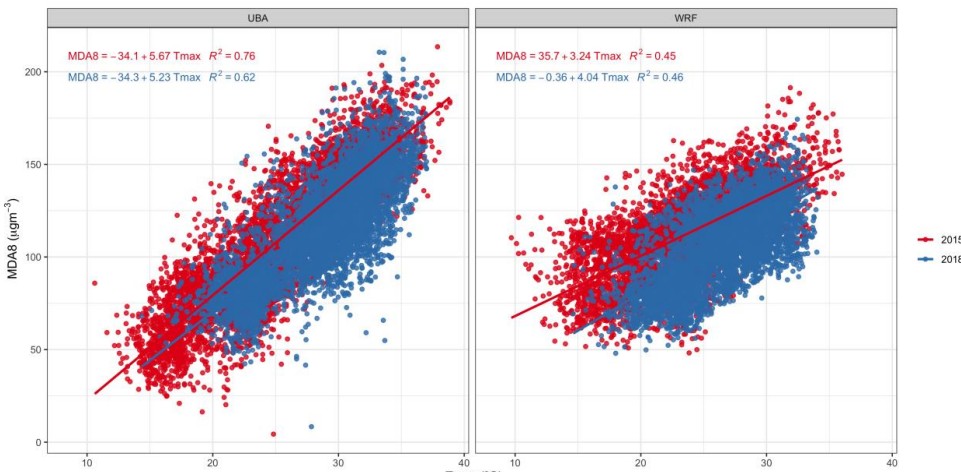


**Figure 4**. Scatter plots showing the MDA8 $O_3$ concentration versus the T2MAX for 2015 (red dots) and 2018 (blue dots) for all analyzed stations. The left panel depicts the observed MDA8 $O_3$ vs ERA5 T2MAX, while the right panel exhibits the modeled MDA8 $O_3$ and T2MAX. The solid lines are the lines of best fit.


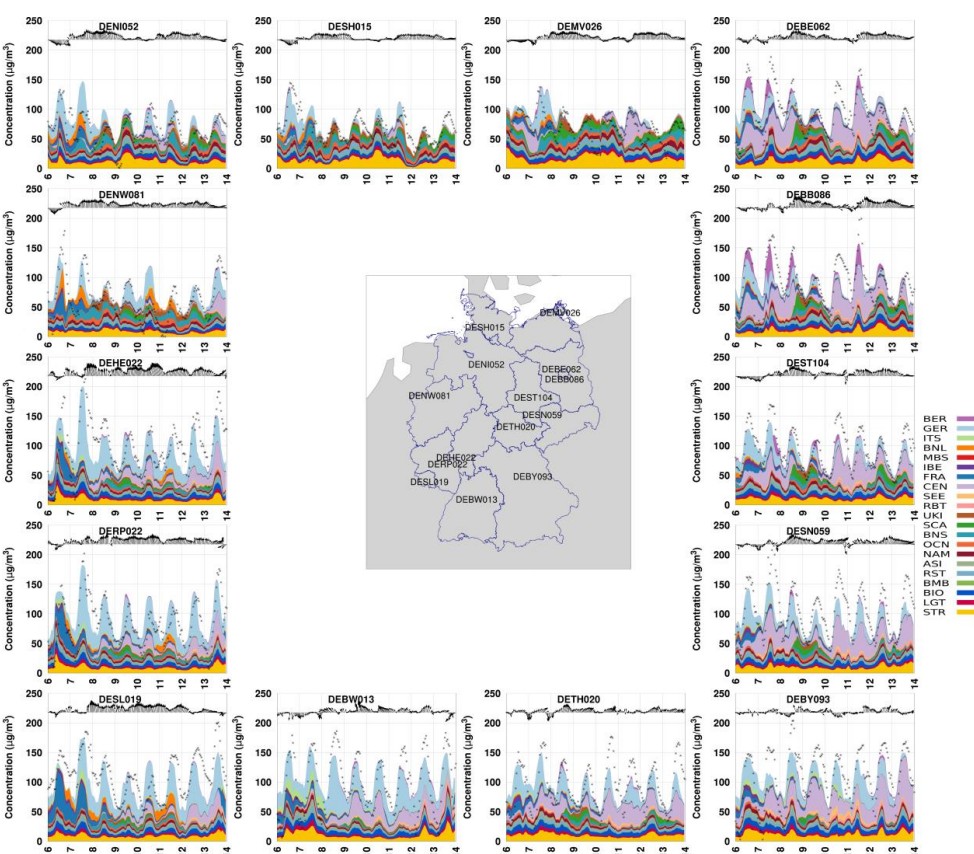

**Figure 5.** Contribution of regional NO$_x$ sources to hourly O$_3$ concentrations of local and other European sources, HTAP2 source regions, and other global source types at each station during the 6 – 13 August 2015 period. In addition, the vectors along the top of each panel represent the calculated wind speed and direction at 10 m above ground level

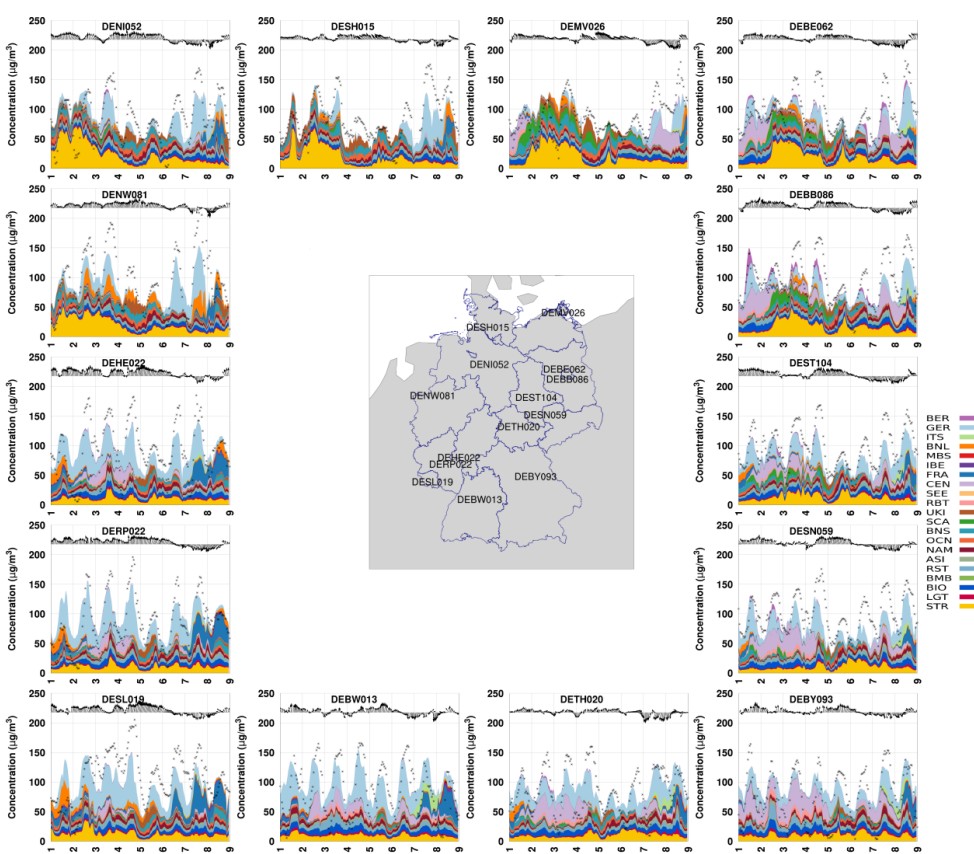

**Figure 6.** Contribution of regional NO$_x$ sources to hourly O$_3$ concentrations of local and other European sources, HTAP2 source regions, and other global source types at each station during the 1 – 8 August 2018 period. In addition, the vectors along the top of each panel represent the calculated wind speed and direction at 10 m above ground level



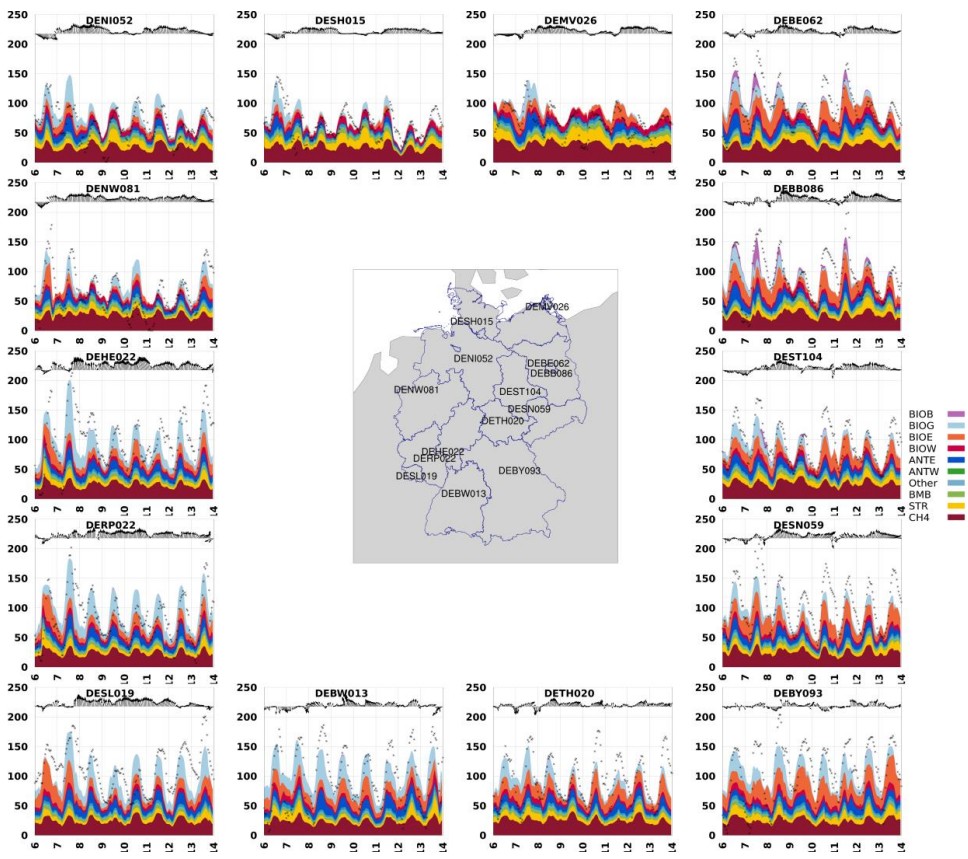

**Figure 7.** Contribution of regional VOC sources to hourly $O_3$ concentrations of local and other European sources, HTAP2 source regions, and other global source types at each station during the 6 – 13 August 2015 period. In addition, the vectors along the top of each panel represent the calculated wind speed and direction at 10 m above ground level




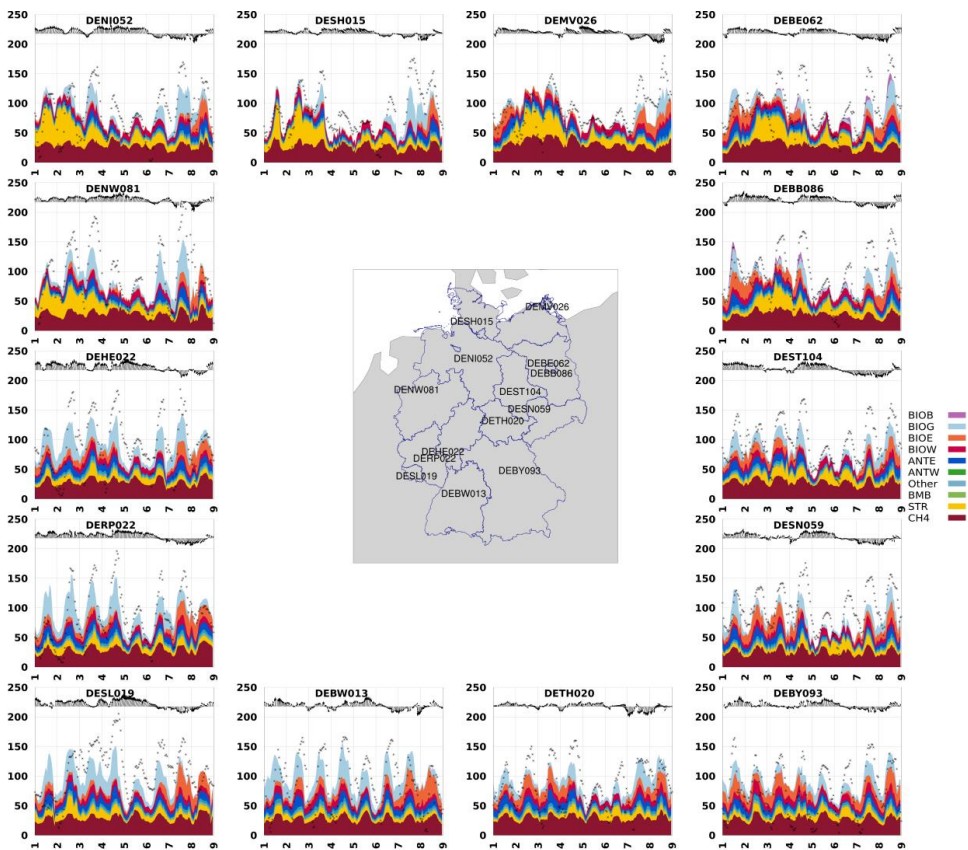

**Figure 8.** Contribution of regional VOC sources to hourly O$_3$ concentrations of local and other European sources, HTAP2 source regions, and other global source types at each station during the 1 – 8 August 2018 period. In addition, the vectors along the top of each panel represent the calculated wind speed and direction at 10 m above ground level


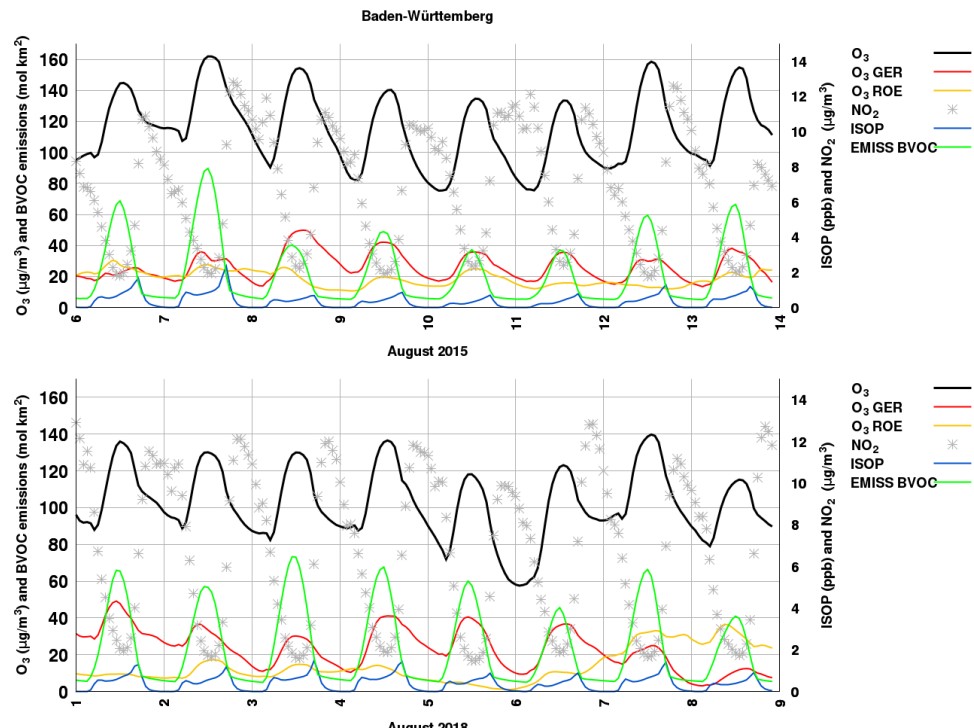

**Figure 9**. Diurnal variation of total ozone and its German and European components together with the total BVOC

emissions and isoprene concentration for 6 – 13 August 2015 (upper panel) and 1 – 8 August 2018 (lower panel)