# Peer review of "Attribution of surface ozone to $NO_x$ and VOC sources during two different high ozone events"

_Atmospheric Chemistry and Physics, 2022_

## Author Comment (AC1)

Response to Referees' comments on the ACPD manuscript "Attribution of surface ozone to  $NO_x$  and VOC sources during two different high ozone events" by Lupaşcu et al.

Dear Referees, Dear Editor,

We would like to thank both reviewers for their positive comments and their time and effort in reviewing the manuscript. We have carefully revised the manuscript according to their comments. Here, we provide our responses. In each case, we have copied the referees' comments and our responses are provided in blue and our changes to the manuscript are in highlighted in red.

**Response to Anonymous Referee #1 comments**

The manuscript presents the high O3 episodes in two heatwaves in Germany, attributing the O3 levels to NOx and VOC sources within and outside Germany using the TOAST system in the WRF-Chem model. The manuscript is very well written, well-structured, and easy to navigate. The tables and figures are relevant and to the point, well-supporting the discussion of the results. There are few points that should be addressed before the manuscript can be published in ACP.

Thank you very much for the positive comments. To address your remaining concerns, we revised the paper and addressed all the concerns raised by responding to individual comments below. Our responses are in blue. Any additions to the manuscript are in red text.

The emissions are based on the TNO-MACC III for the year 2015. Are the emissions used for the 2018 simulation kept the same or somehow scaled? Information on 2018 anthropogenic emissions is lacking.

To perform these simulations we used the same inventory. We updated the sentence as it follows "Anthropogenic emissions (based on the year 2015) of CO,  $NO_x$ ,  $SO_2$ , NMVOCs, PM10, PM2.5, and  $NH_3$  are obtained from the TNO-MACC III emissions inventory (Kuenen et al., 2014) and used for both simulated years."

There is no mention on the dust emissions or if dust is used at all in the simulations, ie from the boundaries. This might be important to consider since some periods are associated with southerlies with pressure systems starting from Africa. Dust can alter the radiation in the atmosphere, which can impact the chemical reactions such as O3 formation. Using an onlinecoupled model has the advantage to account for these feedbacks and not using dust emissions or particles from boundaries can partly be a reason for biases in temperature and O3 levels.

We did not mention the dust since our chemical-tagged chemical mechanisms are not coupled with the aerosol module. However, our results showed that WRF-Chem overestimated the relative humidity and precipitations which likely contributed to the biases in temperature and O3. Hence, following the comments from both Reviewers 1 and 2, we have added the following paragraphs in section 4.2 "Otero et al (2016) showed that in summer, apart from the temperature that is the main driver that dominates high ozone levels, relative humidity plays a negative effect on ozone levels. Hence, the high modeled relative humidity (see Tables 2 and 3) together with overestimated shallow precipitation (see Figures S4 and S5) suggests a reduction in ozone production due to enhanced cloudiness. As noted in Section 4.1, the overestimation of simulated wind speed could enhance the horizontal mixing of ozone and emitted pollutants in our simulations. It is well known that the ground-level ozone depends, amongst other factors, on the level of ozone precursors, such as biogenic VOCs, especially isoprene, and anthropogenic NOx. Unfortunately, the isoprene concentrations, one of the main VOC precursors of ozone that reacts by the high-NOx pathway to produce ozone, are not monitored at the German network sites, so we can't assess the model's ability to reproduce this pollutant. Fast et al. (2014), using the measurements collected in May – June 2010 during the Carbonaceous Aerosol and Radiative Effects Study (CARES) and the California Nexus of Air Quality and Climate Experiment (CalNex), showed that even the model reproduces the temporal isoprene variation, still, daytime mixing ratios of isoprene are usually a factor of 2 too low. Moreover, the underestimation of the modeled surface temperature could lead to an underestimation of isoprene emissions. Guenther et al. (2012) showed that changes in meteorology could lead to a change of ~15% in isoprene and terpene emission fluxes. Thus, the lack of isoprene and the underestimation of predicted NOx (see Tables 4 and 5) could also explain the underestimation of MDA8 O3."

It would be good to describe why the NOx and VOC attribution simulations use different sets of source regions. Is it a simplification based on the complexity or computationally as VOCs included many lumped species?

Generally, the increase in chemical complexity associated with VOC-tagged simulations leads to an increased number of transported tracers which increased the simulation cost. This information is included in Section 2.1 now as follows "The number of explicitly tagged anthropogenic source regions is reduced in the VOC tagged simulation due to the higher computational requirements of the VOC tagging compared with NOx tagging. Moreover, as noted by Butler et al. (2018, 2020a) the contribution of anthropogenic VOC emissions to the total ozone during summer is reduced compared to the contribution of anthropogenic NOx emissions."

Temperature plot in Figure 1 does not show overestimations at nighttime as suggested in the manuscript (Line 216).

Thank you for spotting this, we corrected it.

Figures 5-8. Add that the dots represent the observed O3 concentrations. Thank you for this, we corrected it.

In general, it would be beneficial if the contributions are not only given in absolute terms but also in relative terms (percentages) so the reader can see if the impact is large or small. Thank you for this suggestion, we added the relative contribution throughout the text.

**Response to Anonymous Referee #2 comments**

The manuscript is written in a clear language and well structured. The relevant literature is presented in the introduction. The limitations of the approach are described and discussed. The results are presented and discussed with reference to previous studies. However, the figures reporting the results should be improved because are difficult to visualize. A discussion about possible steps to deal with the model underestimation compared to observations would enhance the impact of the study.

Thank you very much for the positive comments. To address your remaining concerns, we revised the paper and addressed all the concerns raised by responding to individual comments below. Our responses are in blue. Any additions to the manuscript are in red text.

Specific comments Line 70: OH radical We have fixed it.

Line 75: There are two Butler et al. citations in the reference list. The authors should distinguish them with a letter.

Thank you for spotting this, we corrected the references throughout the text.

Line 178: Please clarify the abbreviations of the VOC regions used in Figures 7 and 8 Thank you for pointing this out, we added the abbreviation of each VOC source. Line 210: Please, explain what temporal resolution is used for the comparison (hourly, daily, other?)

We have used hourly data for evaluation, and we added this information in the revised text at Line 214.

Line 228: Please, comment if there are significant differences in the performance indicators between urban and rural sites.

Following the Reviewer's comment, we also evaluate the modeled gas-phase concentration at "rural" and "surburban and urban background" stations. Thus, we updated Tables 4 and 5 to account for these changes. We have updated the manuscript on Lines 236-241: "Generally, the NO2 and NOx concentrations are slightly overestimated at the "rural background" stations, and underestimated at "suburban and urban background stations". This behavior has been also noted by Kushta et al. (2019) and Ghermandi et al. (2020). The underestimation of NO2 at urban stations is mostly an effect of the spatially resolved traffic emission totals, while the overestimation at "rural" stations indicates that the NOx could be transported from the cities to the nearby rural areas." and Lines 251-253, "The lower O3 bias at "rural" stations compared to those seen at "suburban and urban background stations" could be a result of an increase in the NO titration effect due to NOx overestimation at rural stations."

**Line 306: More work is needed to understand the causes of the underestimation**

Following the comments from both Reviewers 1 and 2, we have added the following paragraphs in section 4.2 "Otero et al (2016) showed that in summer, apart from the temperature that is the main driver that dominates high ozone levels, relative humidity plays a negative effect on ozone levels. Hence, the high modeled relative humidity (see Tables 2 and 3) together with overestimated shallow precipitation (see Figures S4 and S5) suggests a reduction in ozone production due to enhanced cloudiness. As noted in Section 4.1, the overestimation of simulated wind speed could enhance the horizontal mixing of ozone and emitted pollutants in our simulations. It is well known that the ground-level ozone depends, amongst other factors, on the level of ozone precursors, such as biogenic VOCs, especially isoprene, and anthropogenic NOx. Unfortunately, the isoprene concentrations, one of the main VOC precursors of ozone that reacts by the high-NOx pathway to produce ozone, are not monitored at the German network sites, so we can't assess the model's ability to reproduce this pollutant. Fast et al. (2014), using the measurements collected in May – June 2010 during the Carbonaceous Aerosol and Radiative Effects Study (CARES) and the California Nexus of Air Quality and Climate Experiment (CalNex), showed that even the model reproduces the temporal isoprene variation, still, daytime mixing ratios of isoprene are usually a factor of 2 too low. Moreover, the underestimation of the modeled surface temperature could lead to an underestimation of isoprene emissions. Guenther et al. (2012) showed that changes in meteorology could lead to a change of  $\sim$ 15% in isoprene and terpene emission fluxes. Thus, the lack of isoprene and the underestimation of predicted NOx (see Tables 4 and 5) could also explain the underestimation of MDA8 O3."

**Line 319: the influence of stomatal resistance on the model performance should be further explored**

Following the Reviewer's suggestion, we have updated the discussion as follows "Rydsaa et al. (2016) showed that WRF-Chem underestimates the modeled nighttime stomatal ozone uptake that leads to too high modeled estimates of ozone concentration in the stable nighttime planetary boundary layer. Also, they showed that during daytime the modeled stomatal conductance is higher than the observations, and thus too low midday modeled ozone concentration. This is also consistent with our simulations (see Fig. 1). As in Rydsaa et al. (2016), Fig. 1 shows a low midday modeled ozone concentration when compared with observation, and high modeled night-time ozone. The stomatal resistance in the Wesely scheme is calculated using the first layer (surface) temperature and solar radiation. Thus, the lasting high 2m temperature in 2018 (maximum of 34.3° C, and median of 21.8° C) compared to those modeled in 2015 (maximum of 36.1° C, and median of 19.6° C) leads to high stomatal resistance, consequently a reduced ozone uptake from vegetation that ultimately leads to an increased modeled surface concentration. Turnipseed et al. (2009) also showed that a major sink for ozone in the canopy is the direct uptake by vegetation through the stomata. Jiang et al. (2018) showed that a reduction of stomatal conductance leads to an increase in leaf temperature, and consequently more isoprene emissions from plants. Using a global model, Gong et al. (2020) found that O3-induced inhibition of stomatal conductance can increase surface O3 by 1.0-1.3 ppbv in western Europe. Visser et al. (2021) also showed that daytime stomatal conductance is overestimated."

**Section 4.3.1: The authors should discuss more the stratospheric ozone intrusion observed in Northern Germany in this section.**

Thank you for this suggestion, however, a full discussion of the stratospheric intrusion is beyond the scope of this paper and it should be discussed in a future paper together with other future analyses needed to determine the stratospheric intrusion such as the analysis of potential vorticity (PV) to diagnose the air mass exchange between troposphere and stratosphere, analysis of vertical profiles of O3 and PV. We updated the text at Line 397 "Future work should examine the stratospheric O3 intrusion and their impact on tropospheric O3 ozone production".

**Line 359: What do you mean by small? Please, quantify**

Following the Reviewer's suggestion, we updated the text as follows "For both episodes, the contribution of German NOx emissions to the ozone concentration is small most of the time in the northern states (from 6.7  $\mu$ g/m3 (6/%) in 2015 and 7.6  $\mu$ g/m3 (8.3%) in 2018 in Mecklenburg-Vorpommern to 9.0  $\mu$ g/m3 (16%) in 2015 and 12.5  $\mu$ g/m3 (13.1%) in 2018 in Brandenburg), whereas the peak ozone events in the south-western and western German stations are mainly driven by German NOx sources (up to 35.31  $\mu$ g/m3 (33.6%) in 2015 and 33.5  $\mu$ g/m3 (33.4 %) in Baden-Württemberg) usually exceeding the total contribution of surrounding source regions.

**Line 368: What to mean by polluted air? polluted with O3 or with other pollutants? We changed the manuscript to " $O_3$ pollution" to avoid any confusion.**

**Line 381: Please, report the share here**

Following the Reviewer's suggestion, we updated the text as follows "Central Europe (CEN) is responsible for a remarkably constant share of ~25 % (ranging from 19 to 30  $\mu$ g/m3) of the total ozone in 2015 at stations in Berlin, Brandenburg, Saxony-Anhalt, Thuringia, Saxony, and Bavaria, while in 2018 it exhibits an erratic contribution to the total O3 concentration (see Figs. 5 and 6)."

Line 427: In the figures can be observed diurnal cycles but not easy for the reader to compare the exact timing between O3 from NOx and VOC

We added in the supplementary material two plots depicting the temporal evolution of NOx and VOC precursors at each of the analyzed stations. The text has been updated to account for this "Comparison of high ozone events in Figs. 6 and 7 with the corresponding events in Figs. 8 and 9 (see also Figs. S8 and S9) show that biogenic VOC contribute to O3 when they react with anthropogenic NOx from nearby sources."

Line 472: In Figure 9 total O3 and the one deriving from GER and ROE are plotted, not clear to which one your refer when you speak of O3 from biogenic sources. Please, be more specific We have updated the text to make it clear that we are referring to German and other European biogenic sources.

Results: it would be useful to add a figure summarizing the overall results where both absolute and relative source contributions are reported.

Following the Reviewer's suggestion, we added heat map plots that depict the mean absolute and relative contribution of NOX and VOC precursors to total  $O_3$  at each receptor station for

the simulated periods. The text has been updated in Section 4.3 as follows "Figures 5 and S6 summarize the mean absolute and relative contribution of NOx and VOC precursors to hourly surface O3 for the analyzed periods. We note that when we attribute O3 to NOx, the southern stations (Baden-Württemberg, Hesse, Rhineland-Palatinate, and Saarland) show a large contribution from the German source region (up to 35.2  $\mu$ g/m3 (34.4%) in 2015 and 33.4  $\mu$ g/m3 (34.7%) in 2018), while for the remaining station the German NOx emissions were not seen as a dominant source, Central Europe source region being one of the most important contributors to the total ozone (up to 30.4  $\mu$ g/m3 (29.1%) in 2015 and 19.7  $\mu$ g/m3 (22%) in 2018). When we attribute O3 to VOC, we note that CH4 is the most significant contributor to the total O3 (up to 31.21  $\mu$ g/m3 (36.3%) in 2015 and 31.5  $\mu$ g/m3 (37.2%) in 2018), followed by German and European BVOCs emissions. Further, we will analyze the hourly variation in O3 concentration to examine the impact of variation in anthropogenic and biogenic emissions, meteorology and long-range transport".

Figures 5 to 8: it is a good idea to combine the graphs of all the German regions in one single figure. However the details are hardly visible and the text/ scales are too small. It is necessary to find a way to summarize the info. My suggestion is to show only the average values for each region in the combo figure and to connect the plots with a line to the relevant region to ease the visualization (reading codes is awkward).

Using average values for each region will reduce the complementary information we have in our temporal variation plots and we have decided to leave them in their current form. However, we increased the readability of the plots by increasing the font size for the station's names, axes, and legends and removing the information on the axis for most of the panels. We have also added arrows to connect the panels to the corresponding locations on Germany's map.

Conclusions: include a statement on what actions should be taken to deal with the overall model underestimation of observations.

Following the Reviewer's suggestion, we update the last paragraph as follows: "Overall, this study provides useful findings on how emissions from local and remote sources influence the predicted O3 and MDA8 O3 during two high ozone episodes. Biogenic VOC emissions as well as the NOx emitted in nearby regions enhance the O3 production during episodes of higher temperatures. Given the high importance of biogenic VOC in determining the peak ozone concentrations, the lack of VOC measurements for evaluation of the modeled VOCs is another source of uncertainty in modeled ozone production. Previous studies highlighted that the model strongly underestimates the isoprene concentrations leading to an underestimation of the total ozone concentration, which might be the case for our study. It is noteworthy that, apart from modeled BVOCs, other factors such as anthropogenic emissions can influence the

oxidation mechanism of the BVOC used in our simulation. Thus, the use of a highly resolved emissions inventory and additional VOC measurements, including biogenic VOC are necessary to improve our understanding of how well the modeled ozone precursors are simulated, consequently, the total O3 concentrations."

**References**

Fast, J. D., Allan, J., Bahreini, R., Craven, J., Emmons, L., Ferrare, R., Hayes, P. L., Hodzic,
A., Holloway, J., Hostetler, C., Jimenez, J. L., Jonsson, H., Liu, S., Liu, Y., Metcalf, A.,
Middlebrook, A., Nowak, J., Pekour, M., Perring, A., Russell, L., Sedlacek, A., Seinfeld, J.,
Setyan, A., Shilling, J., Shrivastava, M., Springston, S., Song, C., Subramanian, R., Taylor, J.
W., Vinoj, V., Yang, Q., Zaveri, R. A., and Zhang, Q.: Modeling regional aerosol and aerosol
precursor variability over California and its sensitivity to emissions and long-range transport
during the 2010 CalNex and CARES campaigns, Atmos. Chem. Phys., 14, 10013–10060,
https://doi.org/10.5194/acp-14-10013-2014, 2014.

Guenther, A. B., Jiang, X., Heald, C. L., Sakulyanontvittaya, T., Duhl, T., Emmons, L. K., & Wang, X. (2012). The Model of Emissions of Gases and Aerosols from Nature version 2.1 (MEGAN2.1): An extended and updated framework for modeling biogenic emissions. Geoscientific Model Development, *5*(*6*), 1471–1492. https://doi.org/10.5194/gmd-5-1471-2012

Jiang, X., Guenther, A., Potosnak, M., Geron, C., Seco, R., Karl, T., Kim, S., Gu, L., Pallardy, S.: Isoprene emission response to drought and the impact on global atmospheric chemistry, *Atmospheric Environment*, 183, pp. 69-83., doi: 10.1016/j.atmosenv.2018.01.026, 2018

Gong, C., Lei, Y., Ma, Y., Yue, X., and Liao, H.: Ozone–vegetation feedback through dry deposition and isoprene emissions in a global chemistry–carbon–climate model, Atmos. Chem. Phys., 20, 3841–3857, https://doi.org/10.5194/acp-20-3841-2020, 2020.

Visser, A. J., Ganzeveld, L. N., Goded, I., Krol, M. C., Mammarella, I., Manca, G., and Boersma, K. F.: Ozone deposition impact assessments for forest canopies require accurate ozone flux partitioning on diurnal timescales, Atmos. Chem. Phys., 21, 18393–18411, https://doi.org/10.5194/acp-21-18393-2021, 2021.

Ghermandi, G.; Fabbi, S.; Veratti, G.; Bigi, A.; Teggi, S. Estimate of Secondary NO2 Levels at Two Urban Traffic Sites Using Observations and Modelling. *Sustainability* **2020**, *12*, 7897. https://doi.org/10.3390/su12197897

Kushta J., Georgiou G.K., Proestos Y., Christoudias T., Thunis P., Savvides C., Papadopoulos C., Lelieveld J.: Evaluation of EU air quality standards through modeling and

the FAIRMODE benchmarking methodology. *Air Qual Atmos Health* **12,** 73–86 (2019). https://doi.org/10.1007/s11869-018-0631-z

---

## Author Response (AR2)

**Comments to the author**:

In addition to the report already available, I have received via email a note from the other original reviewer, indicating that all comments have been addressed.

I have a few suggested word changes for the authors to consider (line numbers refer to the 'track changes' version:

The authors want to thank the editor, John Orlando, and both reviewers for their time and effort in reviewing the revised version of the manuscript.

The answers to specific questions/suggestions are addressed below. All Editor's comments are given in black, and replies in blue.

Line 29: Change the first 'understand' to 'demonstrate'.
We corrected it, thank you.

Line 102: Delete the 's' from 'overpredicts'
We corrected it, thank you.

Line 289: 'precipitation' instead of 'precipitations' ? There may be other occurrences of this as well.
Deleted it

Line 328: 'suggest' instead of 'suggests'
We corrected it, thank you.

Line 393: "ozone-rich" might be better?
Changed to "air masses that contain very high ozone concentrations"

Line 470: subscript on O3.
We corrected it, thank you.